# The Yield, Chemical Composition, and Antioxidant Activities of Essential Oils from Different Plant Parts of the Wild and Cultivated Oregano (*Origanum vulgare* L.)

**Zoran Ilić** [1,*], **Ljiljana Stanojević** [2], **Lidija Milenković** [1], **Ljubomir Šunić** [1], **Aleksandra Milenković** [2], **Jelena Stanojević** [2] **and Dragan Cvetković** [2]

1    Faculty of Agriculture, University of Priština in Kosovska Mitrovica, 38219 Lešak, Serbia
2    Faculty of Technology, University of Niš, 16000 Leskovac, Serbia
*    Correspondence: zorans.ilic@pr.ac.rs; Tel.: +381-638014966

**Abstract:** The present study focuses on the yield, chemical composition, and antioxidant activity of essential oils from different parts (flowers or leaves/stems) of cultivated plants grown under pearl shade nets with a 40% shaded index or in nonshaded plants and wild-grown oregano. The chemical composition of isolated essential oils was determined by GC/MS and GC/FID. Antioxidant activity was determined using the DPPH assay. The highest yield of oregano essential oils (OEOs) was obtained in cultivated shaded plants (flowers) at 0.35 mL/100 g p.m., in contrast to nonshaded plants (flowers), where the yield of OEOs was low (0.21 mL/100 g p.m.). Qualitative and quantitative analyses of the OEOs identified 16–52 constituents that varied with origin and plant organs. The oxygenated sesquiterpene caryophylleneoxide (7.4–49.9%) was predominant in all the essential oil samples. Other major constituents were sesquiterpene hydrocarbon-germacrene D (8.4–22.5%) and (E)-caryophyllene (8.5–10.8%), monoterpene hydrocarbon-sabinene (1.6–7.7%), and oxygen-containing monoterpenes-terpinen-4-ol (1.5–7.0%). The plant part has a significant effect on the antioxidant activity of OEOs, while the influenceof modified light under the shade nets is significantly lower. The OEOs from wild flowers showed the highest antioxidant activity, with an $EC_{50}$ value of 4.78 mg/mL. OEOs from cultivated nonshaded plants (flowers) recorded the lowest antioxidant activity with an $EC_{50}$ value of 24.63 mg/mL. The results suggest that the yield and quality of OEOs can be scaled-up by optimizing plant production in comparison with wild-growing plants. The content and quality of OEO can be increased by optimizing its production compared to plants from the spontaneous flora. Adequate cultivation techniques, such as shading, can achieve high-quality oregano yields and better quality parameters in terms of specific OEO components and meet the different requirements of the market and industrial sectors.

**Keywords:** oregano; cultivation; shading; wild plant; essential oils; antioxidant activity

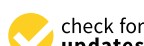



## 1. Introduction

Oregano (*Origanum vulgare* L.) is a well-represented plant and is one of the most commercially important herbs. This plant species is often found in the Mediterranean natural flora, is widely distributed in all Balkan countries [1], and is considered an important resource in the food, pharmaceutical, perfumery, and cosmetic industries [2]. Different plant parts of *Origanum vulgare* subsp. *vulgare* are widely used in Serbia for medicinal and culinary purposes to increase the taste and flavor of foods, or as an ingredient in alcoholic drinks or beverges [3]. Oregano's medicinal properties have been widely used to treat a variety of human diseases, including wounds, coughs, and skin and gastrointestinal issues [4].

Oregano's essential oils (OEOs) have the most intense antioxidant potentials /properties/activities, which are associated with the presence of their phenolic components, thymol and carvacrol.

Due to the content of thymol and carvacrol [5], oregano's essential oils are very good antioxidants and antibacterial protectants in the meat, bread, and cheese industries [6–8]. The essential oil yield and composition are the result of different factors, including genotype [9], environment [10], the geographical position, developmental stage [11], the season of picking, the plant part that is used [12], and cultural practices [13]. The reduction of yield and concentration of phytonutrients in plants is greatly influenced by environmental factors (temperature, relative humidity, light conditions, etc.) [14].

Studies on the cultivation of aromatic plants have found different responses concerning the content and composition of essential oils, according to the light spectrum controlled during cultivation [15–17]. Recently, due to global warming, an increasing number of producers of medicinal plants use shading nets to protect their plants, especially in regions where high temperatures and intense light are present [15].

The EO content of fresh biomass is influenced by the conditions of plant growth, and it increases during the biological cycle, reaching its maximum values at full bloom. Oregano plants with a high thymol content (up to 85%) are not frequent in wild populations. In these cases, carvacrol, the compound responsible for identifying a plant as being of the oregano type, is a minor constituent [18].

The OEO content fluctuates from 0.5 to 2% [19] and up to 7% [20], and its main components are the isomer phenols, carvacrol, and thymol, as well as their precursor monoterpenes p-cymene and γ-terpineneatain a smaller proportion [19]. The most represented components in OEOs from Serbia are: caryophylleneoxide (3.1–1.93%); germacrene D (1.17–2.0%); and (E)-caryophyllene (1.48–1.1%) [17]. Similar to the variation in OEO yields, the number of detected compounds has a wide range. The smallest number of compounds (19) is reported for one Greek population [21], while the biggest one (111) is reported for one Serbian population [17]. In the examined Montenegrin population, 30 compounds were detected [1]. Although several studies have been conducted to practice some agriculture techniques for oregano production [15,16,22–24] to improve essential oil content, physiological mechanisms and biochemical properties are still not clear.

Moving plants from spontaneous flora and growing them in new conditions (cultivated production) involves a number of limitations. In response to global warming, we use shading nets to modify light and imitate natural environmental conditions, creating more favorable conditions for the growth and development of oregano in intensive field production. Growing oregano in shaded conditions improved the quantity and quality of EOs. A possible practical application of this study would be to grow oregano at higher plant densities, or as an intercrop, because it does not require too much light for production.

This research was aimed at investigating the essential oil content, chemical composition, and antioxidant activity of pink flowered oregano (*Origanum vulgare* subsp. *vulgare* L.) obtained from aerial parts (leaves and flowers) of cultivated (shading or nonshading conditions) and wild oregano grown in Serbia.

## 2. Material and Method

### 2.1. Plant Materialand Growing Conditions

The experiment was carried out in an experimental garden in the village of Moravac in South Serbia (21°42′ E, 43°30′ N, altitude 159 m a.s.l. between 2020 and 2021. Oregano (*Origanum vulgare* L. subsp. *vulgare* L.) seed was distributed by the Suba Seed Company SPA Srl. Longiano, Italy.

Growing oregano is best on flat or slightly undulating land. Deep plowing is required. To improve soil texture and fertility, ten metric tons of manure per hectare should be applied during soil preparation. The soil should be cultivated two or three times to allow the manure to break down and, prevent the growth of weeds, and eliminate grubs and fungi. In addition to stable manure, we added more than 300 kg of triple calcium superphosphate. The sowing of oregano was performed on 20 May, and after germination, the crop was thinned so that a plant density of 50 plants/m$^2$ was achieved. Oregano was used to determine whether shading conditions (plants covered by pearl nets with a shade index

of 50%, Polysack, Israel) could improve essential oil composition and antioxidant activity in plants. Combinations of plant shading treatments and unnetted control treatments were replicated three times in a split-plot design. From June to the end of August, the oregano was covered with shading nets mounted on a supporting structure above the plants. The oregano plants were harvested in the middle of August, at the commercial maturity stage (ten to twelve weeks after germinating). After summer cutting, the plants may have vegetative re-growth until October. To ensure this re-growth, it is necessary to irrigate the crop during the dry period, particularly from June to September. It is best to harvest the oregano in the morning, after the dew has dried. When the leaves dry, they can be easily separated from the stems.

The wild-grown oregano at the bottom of the Tara Mountains was collected at the flowering stage to determine the chemical composition and biological activities of essential oils from wild-grown oregano. When the plants were in the flowering stage in 2020 and 2021, harvest was completed in late August.

### 2.2. Clevenger-Hydrodistillation

Disintegrated and homogenized plant material was used for essential oil isolation by Clevenger-type hydrodistillation, with a hydromodulus (ratio of plant material: water) of 1:10 m/V for 120 min. During the distillation, the volume of separated essential oils was read in the measuring Clevenger's tube after 15–120 min, and was monitored depending on the yield of the essential oil over the time. Essential oil isolated from each sample was separated from the measuring tube after distillation, dried over hydrous sodium sulfate and stored in dark bottles in a refrigerator at +4 °C.

### 2.3. Gas Chromatography/Massspectrometry (GC/MS) and Gas Chromatography/Flameionization Detection (GC/FID) Analysis

The details of GC/MS and GC/FID analyses are given in Ilić et al. [16], with those available in the literature [25,26].

### 2.4. Antioxidantactivity(DPPH Assay)

Oregano essential oils were diluted with ethanol, and a series of different concentrations were created (0.002–0.2 mg/mL). The procedure was conducted with two probes. Namely, the absorbance at 517 nm was immediately measured in the first probe, while the absorbance in the second probe was measured after 60 min of incubation at room temperature in the dark (as absorbance of the sample). All other relevant details of the assay used are given in Stanojević et al. [27,28].

## 3. Results and Discussion

### 3.1. Growing Condition

The climatic conditions of southern Serbia are very favorable for the production of oregano. In order to create adequate microclimatic conditions for better production and quality of oregano plants, shading nets were used. Photosynthetically active radiation (PAR) was significantly lower under pearl nets with 40% shading (1100 mmol $s^{-1}m^2$) compared to the control-open field condition (2242 μmol $m^{-2}s^{-1}$). The use of shading nets, solar radiation is also reduced from 874 $Wm^2$ in the open field to 459 $Wm^2$ behind the nets (Table 1).

*Origanum vulgare* L. is the species with the greatest genetic diversity in the genus Origanum [29].The cultivation of wild oregano species requires a good knowledge of the biology of the plant. The reason that such microclimatic conditions were created by the use of shading nets for the cultivation of wild oregano is because of their better adaptation to new conditions while imitating the conditions of the environment from which they originate. This primarily refers to the conditions of light and shading in the natural habitat where the plants grew before their transfer and cultivation in the field.

**Table 1.** Influence of shading on growing environment (average day in July).

| Time (h) | PAR * (μmolm$^{-2}$s$^{-1}$) | | Solar Radiation (Wm$^{-2}$) | | Temperature °C | | Relative Humidity% | |
|---|---|---|---|---|---|---|---|---|
| | Nonshading | Shading Reduction% | Nonshading | Shading | Nonshading | Shading Reduction % | Nonshading | Shading Reduction % |
| 6:00 | 182.5 | 31.2 | 162.5 | 40.5 | 16.7 | 0.0 | 74.7 | −4.1 |
| 9:00 | 1325.6 | 46.0 | 513.8 | 281.0 | 24.7 | −0.4 | 71.8 | 0.0 |
| 12:00 | 2242.2 | 49.1 | 874.5 | 459.5 | 31.4 | −2.2 | 47.3 | −2.1 |
| 15:00 | 1684.1 | 51.9 | 790.5 | 351.0 | 31.5 | −3.4 | 48.2 | −1.2 |
| 18:00 | 672.0 | 53.9 | 375.5 | 90.9 | 28.3 | −1.0 | 50.4 | −0.2 |

* PAR-Photosynthetically active radiation.

### 3.2. Essential Oil Yield

Despite the fact that oregano is the most widely used and represented plant species on the market, there is no accurate data on the quality of oregano or the influence of certain agrotechnical methods in the production of oregano on the yield and quality of essential oils. The collection of oregano plants from natural flora is characterized by instability in the quality as well as a negative impact on the environment. For these reasons, new high-yielding populations promote cultivation instead of natural production.

The highest yield of oregano essential oils (OEOs) obtained after 120 min of hydrodistillation in cultivated shaded plants (flowers) was 0.35 mL/100 g p.m., in contrast to nonshaded plants (flowers), where the yield of OEOs was the lowest (0.21 mL/100 g p.m.). Plants covered by shade nets obtained higher EO content than nonshaded plants. The amount of OEOs varied between the parts of the plant, ranging from 0.33–0.35% in flowers to 0.26–0.32% in stems and leaves (Table 2).

**Table 2.** Yield of essential oil from different parts of the cultivated (shaded and nonshaded) and wild oregano (*Origanum vulgare* subsp. *vulgare* L.) obtained after 120 min of hydrodistillation (hydromodule 1:10 *m/v*).

| Sample | Essential Oil Yield, mL/100 g p.m. |
|---|---|
| Oregano | |
| Cultivated nonshaded (stemsandleaves) | 0.31 ± 0.015 a* |
| Cultivated shaded (stems and leaves) | 0.32 ± 0.013 a |
| Cultivated nonshaded (flowers) | 0.21 ± 0.010 b |
| Cultivated shaded (flowers) | 0.35 ± 0.011 a |
| Wild (stems and leaves) | 0.26 ± 0.018 b |
| Wild (flowers) | 0.33 ± 0.009 a |

* Values followed by different letters are significantly different at $p < 0.05$.

The yield of oregano essential oil depending on the time of hydrodistillation is shown in Figure 1.

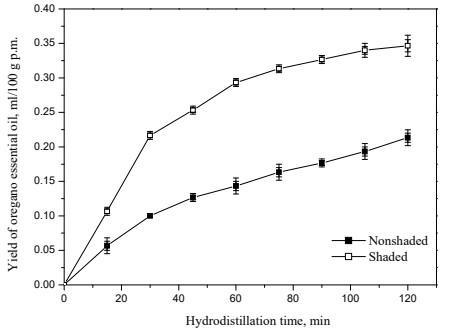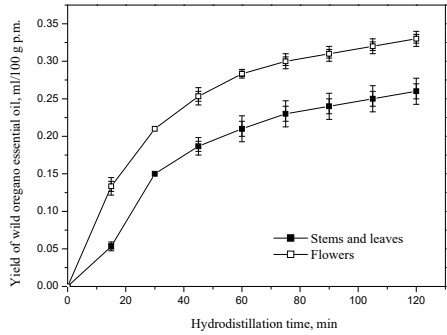

**Figure 1.** Extraction yield (mL essential oils/100 g dried herb) as a function of time for hydrodistillation of essential oils from cultivated (shaded and nonshaded) and wild oregano plants.

This study clearly showed that OEOs increased in all above-ground plant organs from shaded plants. The results from this study are in agreement with Ilić et al. [16], who showed that the yield of essential oils (EOs) from shaded plants showed higher EO content than nonshaded plants. The highest yields of essential oil from the *Melissa officinalis* L. are achieved when the plants are covered with blue nets [30], while red nets achieve the best effects in the production of basil [15]. It is similar in the production of thyme, marjoram, and oregano, where the content of essential oils was higher in plants covered with pearl nets [17]. The content of essential oil compounds of European *O. vulgare* ranged between 0.03% and 4.6% [31].

In general, oregano as a species contains less EO (0.27–0.32%) than other medicinal plants from this family, so the yield of essential oils in oregano is much lower than that of thyme (2.32–2.57%) and marjoram (1.51–1.68%) [17]. An even smaller EOs production oforegano plants is recorded in lemon balm plants, where the content is quite low compared to other plants in the Lamiaceae family [16].

Regarding *O. vulgare* subsp. *vulgare*, all examined populations from Montenegro were rather poor in EO [1]. The highest OEOs yield was observed in the Mediterranean population (1.2%), while others had lower yields (0.7–0.9%). The populations of oregano in southern Albania had a higher amount of essential oil (3.45%) compared with those in the northern part (0.1%) [32]. The higher content of essential oil could be explained by the dryness, the thermal efficiency of the habitat, and as well as by the lower altitude [1]. The essential oil content of oregano from Iran showed a wide variability, ranging from 0.12% to 1.76% (*v/w*), correlating to the chemical profile [33]. The yield of the EOs from Iranian *O. vulgare* L. subsp. *vulgare* (thymol chemotype) was 0.5%, with a pale yellow color and a pungent odor [34].

Additionally, the concentration of oil depends on the population and the climatic conditions. The results of Goyal et al. [35] showed great variation in its EO content depending on geographical location and altitude of cultivation in India (Himalayan region).

While EO yields in *Origanum vulgare* subsp. *vulgare* rarely exceeded 1% [36,37], they are usually greater than 2.5% [19] in *Origanum syriacum* from Turkey (2.9–3.5%), according to Arslan [38]. It is explained by the fact that *O. vulgare* subsp. *vulgare* has fewer gland hairs and is essential oil deficient [39].

Oregano essential oil content depends, among other things, on the plant part and the stage of plant development. According to Putievsky et al. [40], the essential oil content of oregano is higher in the full bloom stage than in the stage of flowering. Oregano plants from central Poland contained, depending on the plant's developmental phase, air-dried samples 0.20–0.58% or 0.35–0.87% of essential oil from eastern Poland [36,41].

Results from this study are in agreement with our research, where the yield of OEOs is slightly higher in flowers than in leaves/steams.

*3.3. Essential Oil Composition*

Oregano essential oils' (OEOs) composition is conditioned by the geographic region of the origin, the method of production, the time of harvest, the extraction method, etc. Qualitative and quantitative analyses of the essential oil identified 16–52 constituents that varied with plant origin and plant organs. The oxygenated sesquiterpene-caryophyllene oxide (7.4–49.9%) was predominant in all the essential oil samples. Other major constituents were sesquiterpene hydrocarbons-germacrene D (8.4–22.5%), (E)-caryophyllene (8.5–10.8%), monoterpene hydrocarbons-sabinene (1.6–7.7), and oxygen-containing monoterpenes-terpinen-4-ol (1.5–7.0%).

Among monoterpenes, hydrocarbons ranged between 2.4 and 23.7%, whereas oxygenated monoterpenes (i.e., monoterpenoids) showed a range of 4.8–22.4%. Sesquiterpene hydrocarbons were always represented at the highest levels (23.8–44.4%), while the oxygenated sesquiterpene fraction was 16.1–65.4%. Aromatic compounds were present in quite small amounts (0.4–2.3%).

Forty-nine compounds of OEOs were identified in the steams/leaves of the shaded plants. Caryophyllene oxide (20.4%), sabinene (10.5%), and germacrene D (8.4%) were the most dominant compounds. The following components are somewhat less represented: (E)-caryophyllene (6.4%), (Z)-β-ocimene (6.3%), and terpinen-4-ol (3.8%), (Table 3).

**Table 3.** Chemical composition of essential oil isolated from shaded and nonshaded oregano (stem, leaves).

| N⁰ | t ret., min | Compound | RI$^{exp}$ | RI$^{lit}$ | Method of Identification | c% | |
|---|---|---|---|---|---|---|---|
| | | | | | | *shaded* | *nonshaded* |
| 1. | 6.70 | α-Thujene | 924 | 924 | RI, MS | tr | tr |
| 2. | 6.92 | α-Pinene | 932 | 932 | RI, MS | tr | 0.2 |
| 3. | 8.19 | Sabinene | 973 | 969 | RI, MS | 7.7 | 10.5 |
| 4. | 8.28 | β-Pinene | 976 | 974 | RI, MS, Co-I | tr | 0.5 |
| 5. | 8.63 | 1-Octen-3-ol | 977 | 974 | RI, MS | 2.3 | 2.4 |
| 6. | 8.71 | Myrcene | 980 | 988 | RI, MS | 1.9 | 1.7 |
| 7. | 9.12 | 3-Octanol | 994 | 988 | RI, MS | 0.4 | 0.4 |
| 8. | 9.67 | α-Terpinene | 1010 | 1014 | RI, MS | 0.4 | 0.4 |
| 9. | 10.06 | p-Cymene | 1021 | 1020 | RI, MS | 1.5 | 2.3 |
| 10. | 10.18 | 1,8-Cineole | 1025 | 1026 | RI, MS, Co-I | 1.7 | 2.1 |
| 11. | 10.45 | (Z)-β-Ocimene | 1030 | 1032 | RI, MS | 7.7 | 6.3 |
| 12. | 10.84 | (E)-β-Ocimene | 1041 | 1044 | RI, MS | 2.9 | 2.2 |
| 13. | 11.27 | γ-Terpinene | 1054 | 1054 | RI, MS | 1.5 | 1.6 |
| 14. | 11.90 | *cis*-Sabinenehydrate | 1069 | 1065 | RI, MS | 0.5 | 0.6 |
| 15. | 12.41 | Terpinolene | 1083 | 1086 | RI, MS | 0.2 | 0.3 |
| 16. | 12.55 | *trans*-Linalooloxide(furanoid) | 1086 | 1084 | RI, MS | tr | tr |
| 17. | 12.78 | Rosefuran | 1092 | 1095 | RI, MS | tr | tr |
| 18. | 13.23 | Linalool | 1103 | 1095 | RI, MS, Co-I | 3.8 | 2.6 |
| 19. | 14.15 | *cis*-p-Menth-2-en-l-ol | 1126 | 1118 | RI, MS | tr | tr |
| 20. | 14.34 | p-Mentha-1,5,8-triene* | 1130 | 1139 | RI, MS | tr | - |
| 21. | 14.97 | *trans*-p-Menth-2-en-1-ol* | 1145 | 1136 | RI, MS | tr | - |
| 22. | 15.36 | β-Pineneoxide* | 1155 | 1154 | RI, MS | tr | - |
| 23. | 15.50 | Sabinaketone | 1158 | 1154 | RI, MS | tr | tr |
| 24. | 16.54 | Terpinen-4-ol | 1182 | 1174 | RI, MS | 4.7 | 3.8 |
| 25. | 17.25 | α-Terpineol | 1200 | 1196 | RI, MS | 2.0 | 1.8 |
| 26. | 19.01 | Neral | 1242 | 1235 | RI, MS, Co-I | 0.3 | 2.6 |
| 27. | 20.31 | Geranial | 1273 | 1264 | RI, MS, Co-I | 0.6 | 2.7 |
| 28. | 20.96 | DihydroedulanI | 1288 | 1288 | RI, MS | 0.6 | 0.7 |
| 29. | 22.05 | (2E,4E)-Decadienol | 1314 | 1319 | RI, MS | tr | tr |
| 30. | 24.51 | Piperitenonoxide | 1373 | 1366 | RI, MS | 1.4 | tr |
| 31. | 24.61 | α-Copaene* | 1375 | 1374 | RI, MS | tr | - |
| 32. | 25.02 | β-Bourbonene* | 1385 | 1387 | RI, MS | 2.3 | - |
| 33. | 25.36 | β-Elemene | 1394 | 1389 | RI, MS | tr | 3.6 |
| 34. | 26.51 | (E)-Caryophyllene | 1422 | 1417 | RI, MS | 8.5 | 6.2 |

Table 3. *Cont.*

| N⁰ | *t* ret., min | Compound | RI$^{exp}$ | RI$^{lit}$ | Method of Identification | c% | |
|---|---|---|---|---|---|---|---|
| | | | | | | *shaded* | *nonshaded* |
| 35. | 26.87 | β-Copaene | 1431 | 1430 | RI, MS | 0.4 | 0.6 |
| 36. | 27.44 | Aromadendrene | 1445 | 1439 | RI, MS | tr | tr |
| 37. | 27.92 | α-Humulene | 1457 | 1452 | RI, MS | 1.2 | 0.8 |
| 38. | 28.14 | Alloaromadendrene | 1463 | 1458 | RI, MS | 0.5 | 0.5 |
| 39. | 29.09 | GermacreneD | 1485 | 1484 | RI, MS | 13.5 | 8.4 |
| 40. | 29.64 | Bicyclogermacrene | 1500 | 1500 | RI, MS | 2.6 | 1.0 |
| 41. | 29.78 | α-Muurolene* | 1504 | 1500 | RI, MS | tr | - |
| 42. | 30.03 | (E,E)-α-Farnesene | 1510 | 1505 | RI, MS | 2.6 | 1.5 |
| 43. | 30.40 | γ-Cadinene* | 1520 | 1513 | RI, MS | tr | - |
| 44. | 30.69 | δ-Cadinene | 1528 | 1522 | RI, MS | 1.6 | 1.2 |
| 45. | 33.19 | Caryophylleneoxide | 1592 | 1582 | RI, MS | 18.1 | 20.4 |
| 46. | 33.54 | Salvial-4(l4)-en-l-one* | 1603 | 1594 | RI, MS | tr | - |
| 47. | 34.17 | HumuleneepoxideII | 1618 | 1608 | RI, MS | 1.7 | 1.8 |
| 48. | 34.48 | Alloaromadendreneepoxide | 1629 | 1639 | RI, MS | 0.6 | 0.5 |
| 49. | 35.41 | α-Muurolol* | 1647 | 1644 | RI, MS | 0.3 | - |
| 50. | 35.58 | epi-α-Muurolol | 1650 | 1640 | RI, MS | 1.2 | 1.2 |
| 51. | 36.09 | α-Cadinol | 1662 | 1652 | RI, MS | 2.2 | 2.3 |
| 52. | 37.26 | Amorpha-4,9-dien-2-ol | 1706 | 1700 | RI, MS | 1.5 | 1.7 |
| | | | | | Total identified | 99.5 | 99.2 |
| Grouped components (%) | | | | | | | |
| Monoterpene hydrocarbons (1–4,6,8,11–13,15) | | | | | | 22.3 | 23.7 |
| Oxygen-containing monoterpenes (10,14,16–28,31) | | | | | | 14.2 | 18.7 |
| Sesquiterpene hydrocarbons (30,32–44) | | | | | | 33.2 | 23.8 |
| Oxygenated sesquiterpenes (45–52) | | | | | | 25.6 | 27.9 |
| Aromatic compounds (9) | | | | | | 1.5 | 2.3 |
| Others (5,7,29) | | | | | | 2.7 | 2.8 |

Present only in shaded plants: p-Mentha-1,5,8-triene*; *trans*-p-Menth-2-en-1-ol*; β-Pineneoxide*; α-Copaene*; β-Bourbonene*; α-Muurolene*; γ-Cadinene*; Salvial-4(l4)-en-l-one*; α-Muurolol*. Present only in nonshaded plants: Camphene; Limonene; Perillene; Myrtenal; Borneol; (3Z)-Hexenyl 3-methylbutanoate.

Fifty-two compounds were recognized in steams/leaves from nonshaded plants, mainly sesquiterpene hydrocarbons (33.2%) and oxygenated sesquiterpenes (25.6%). Caryophyllene oxide (18.1%) and germacrene D (13.5%) were the most dominant compounds. The following components were present on a slightly smaller scale: (E)-caryophyllene (8.5%), (Z)-β-ocimene (7.7%) and sabinene (7.7%).

The other most abundant compounds of OEOs from steams/leaves always reached more than 3%. Apart from the content of sabinene, which was higher in nonshaded plants (10.5%), the content of other OEO components, such as (Z)-β-ocimene (7.7%); (E)-caryophyllene (8.5%), and terpinen-4-ol (4.7%), was higher in shaded plants.

Even though no literature data could be found to compare with our EO profile, the characteristics and new compounds of the species were also detected in our study. It is very interesting to point out that some components of essential oils are present only in plants that are shaded (p-mentha-1,5,8-triene; *trans*-p-menth-2-en-l-ol; β-pinene oxide; α-copaene; β-bourbonene; α-muurolene; γ-cadinene; salvial-4(l4)-en-l-one; and α-muurolol),

while some others are present only in nonshaded plants that grow in full light (camphene; limonene; perillene; myrtenal; borneol; (3Z)-hexenyl 3-methylbutanoate).

Fifty-one components were detected in flowers from shaded plants, mainly oxygenated sesquiterpenes (76.1%). Caryophylleneoxide (25.5%), germacrene D (14.5%), and (E)-caryophyllene (10.8%) are the most present components in flower samples from shaded plants, Table 4.

**Table 4.** Chemical composition of essential oils isolated from shaded and nonshaded oregano flowers.

| N° | t ret., min | Compound | RI$^{exp}$ | RI$^{lit}$ | Method of Identification | c% Shaded | Nonshaded |
|----|----|----|----|----|----|----|----|
| 1. | 8.16 | Sabinene | 962 | 969 | RI, MS | tr | |
| 2. | 8.63 | 1-Octen-3-ol | 977 | 974 | RI, MS | 1.9 | |
| 3. | 9.12 | 3-Octanol | 994 | 988 | RI, MS | 0.3 | |
| 4. | 9.67 | α-Terpinene | 1010 | 1014 | RI, MS | tr | |
| 5. | 10.05 | p-Cymene | 1020 | 1020 | RI, MS | 0.4 | |
| 6. | 10.17 | 1,8-Cineole | 1025 | 1026 | RI, MS, Co-I | 0.4 | |
| 7. | 10.43 | (Z)-β-Ocimene | 1030 | 1032 | RI, MS | 1.4 | |
| 8. | 10.84 | (E)-β-Ocimene | 1041 | 1044 | RI, MS | 0.5 | |
| 9. | 11.28 | γ-Terpinene | 1052 | 1054 | RI, MS | 0.5 | |
| 10. | 11.91 | cis-Sabinenehydrate | 1069 | 1065 | RI, MS | 0.6 | |
| 11. | 12.43 | Terpinolene | 1083 | 1086 | RI, MS | tr | |
| 12. | 12.57 | trans-Linalooloxide(furanoid) | 1086 | 1084 | RI, MS | tr | |
| 13. | 13.23 | Linalool | 1103 | 1095 | RI, MS, Co-I | 4.7 | tr |
| 14. | 14.15 | cis-p-Menth-2-en-l-ol | 1126 | 1118 | RI, MS | 0.4 | |
| 15. | 14.97 | trans-p-Menth-2-en-1-ol | 1145 | 1136 | RI, MS | tr | |
| 16. | 15.36 | β-Pineneoxide | 1155 | 1154 | RI, MS | tr | |
| 17. | 15.50 | Sabinaketone | 1158 | 1154 | RI, MS | 0.3 | |
| 18. | 16.16 | Borneol | 1173 | 1165 | RI, MS, Co-I | tr | |
| 19. | 16.54 | Terpinen-4-ol | 1182 | 1174 | RI, MS | 7.0 | 1.5 |
| 20. | 17.25 | α-Terpineol | 1200 | 1196 | RI, MS | 3.1 | 3.3 |
| 21. | 18.58 | (3Z)-Hexenyl3-methylbutanoate | 1232 | 1232 | RI, MS | tr | |
| 22. | 18.74 | Thymol, methylether | 1235 | 1232 | RI, MS | tr | |
| 23. | 19.01 | Neral | 1242 | 1235 | RI, MS, Co-I | tr | |
| 24. | 19.16 | Cuminaldehyde | 1245 | 1238 | RI, MS | tr | |
| 25. | 20.31 | Geranial | 1273 | 1264 | RI, MS, Co-I | tr | |
| 26. | 20.96 | DihydroedulanI | 1288 | 1288 | RI, MS | 0.6 | |
| 27. | 22.05 | (2E,4E)-Decadienol | 1314 | 1319 | RI, MS | tr | |
| 28. | 24.61 | α-Copaene | 1375 | 1374 | RI, MS | tr | |
| 29. | 25.02 | β-Bourbonene | 1385 | 1387 | RI, MS | 2.8 | 1.8 |
| 30. | 25.36 | β-Elemene | 1394 | 1389 | RI, MS | tr | tr |
| 31. | 26.51 | (E)-Caryophyllene | 1422 | 1417 | RI, MS | 10.8 | 7.3 |
| 32. | 26.86 | β-Copaene | 1431 | 1430 | RI, MS | 0.5 | |
| 33. | 27.92 | α-Humulene | 1457 | 1452 | RI, MS | 1.6 | tr |

**Table 4.** *Cont.*

| N° | *t* ret., min | Compound | RI^exp | RI^lit | Method of Identification | c% Shaded | c% Nonshaded |
|---|---|---|---|---|---|---|---|
| 34. | 28.14 | Alloaromadendrene | 1463 | 1458 | RI, MS | 0.6 | |
| 35. | 28.85 | γ-Muurolene | 1481 | 1478 | RI, MS | tr | |
| 36. | 29.09 | GermacreneD | 1485 | 1484 | RI, MS | 14.5 | 12.7 |
| 37. | 29.64 | Bicyclogermacrene | 1500 | 1500 | RI, MS | 2.6 | 3.0 |
| 38. | 29.78 | α-Muurolene | 1504 | 1500 | RI, MS | tr | |
| 39. | 30.03 | (E,E)-α-Farnesene | 1510 | 1505 | RI, MS | 3.3 | |
| 40. | 30.40 | γ-Cadinene | 1520 | 1513 | RI, MS | 0.3 | |
| 41. | 30.69 | δ-Cadinene | 1528 | 1522 | RI, MS | 1.7 | 2.7 |
| 42. | 31.34 | α-Cadinene | 1545 | 1537 | RI, MS | tr | |
| 43. | 32.24 | l-nor-Bourbonanone | 1569 | 1561 | RI, MS | 0.5 | |
| 44. | 33.19 | Caryophylleneoxide | 1592 | 1582 | RI, MS | 25.5 | 49.9 |
| 45. | 33.54 | Salvial-4(l4)-en-l-one | 1603 | 1594 | RI, MS | 0.6 | |
| 46. | 34.17 | HumuleneepoxideIl | 1618 | 1608 | RI, MS | 2.2 | 4.6 |
| 47. | 35.41 | α-Muurolol | 1647 | 1644 | RI, MS | 0.5 | 1.0 |
| 48. | 35.58 | epi-α-Muurolol | 1650 | 1640 | RI, MS | 1.8 | |
| 49. | 36.09 | α-Cadinol | 1662 | 1652 | RI, MS | 3.4 | 5.3 |
| 50. | 36.84 | Germacra-4(15),5,10(14)-trien-1-α-ol | 1694 | 1685 | RI, MS | 1.0 | |
| 51. | 37.26 | Amorpha-4,9-dien-2-ol | 1706 | 1700 | RI, MS | 1.9 | |
| | | | | | Total identified | 98.2 | 99.8 |

| Grouped components (%) | Shaded | Nonshaded |
|---|---|---|
| Monoterpene hydrocarbons (1,4,7–9,11) | 2.4 | |
| Oxygen–containing monoterpenes (6,10,12–20,23,25,26) | 17.1 | 4.8 |
| Oxygenated sesquiterpenes (28–42) | 38.7 | |
| Sesquiterpene hydrocarbons (4–10) | | 27.5 |
| Oxygenated sesquiterpenes (43–51) | 37.4 | 65.4 |
| Aromatic compounds (5,22,24) | 0.4 | 2.1 |
| Others (2,3,21,27) | 2.2 | |

Shaded plants presented 35 compounds more than nonshaded plants. Present only in nonshaded plants: isomyristicin (2.1); eudesma-4(15),7-dien-1β-ol(4.6).

Terpinen-4-ol (7%), linalool (4.7%), (E,E)-α-farnesene (3.3%), α-cadinol (3.4%), and α-terpineol (3.1%) are represented to a greater extent than 3% in OEOs.

Some components of essential oils are present only in plants that are shaded (35 compounds), while some others are present only in nonshaded plants that grow in full light: isomyristicin (2.1); eudesma-4(15), 7-dien-1β-ol (4.6).

The OEOs of flowers from nonshaded plants were detected with only sixteen components, mainly oxygenated sesquiterpenes (65.4%). Caryophyllene oxide was the most dominant component at 49.9%. Germacrene D also participated at 12.7%. Most of them are present only in nonshaded plants in an open field. Thus, (E)-caryophyllene (7.3%); humulene epoxide II (4.6%); eudesma-4(15), 7-dien-1 β-ol (4.6%), and bicyclogermacrene (3%) are present only in OEOs of flowers from nonshaded plants (Table 5).

**Table 5.** Chemical composition of essential oil isolated from wild oregano (stems/leaves and flowers).

| N°. | *t* ret., min | Compound | RI$^{exp}$ | RI$^{lit}$ | Method of Identification | c% | |
|-----|-----|-----|-----|-----|-----|-----|-----|
| | | | | | | Steam/Leaves | Flowers |
| 1. | 6.70 | α-Thujene | 924 | 924 | RI, MS | tr | tr |
| 2. | 6.93 | α-Pinene | 932 | 932 | RI, MS | tr | tr |
| 3. | 7.41 | Camphene | 947 | 946 | RI, MS | tr | tr |
| 4. | 8.16 | Sabinene | 973 | 969 | RI, MS | 1.6 | 4.5 |
| 5. | 8.28 | β-Pinene | 976 | 974 | RI, MS, Co-I | tr | 0.5 |
| 6. | 8.64 | 1-Octen-3-ol | 977 | 974 | RI, MS | 2.8 | 2.0 |
| 7. | 8.71 | Myrcene | 980 | 988 | RI, MS | 1.1 | 1.2 |
| 8. | 9.12 | 3-Octanol | 994 | 988 | RI, MS | 0.6 | 0.3 |
| 9. | 9.67 | α-Terpinene | 1010 | 1014 | RI, MS | tr | 0.5 |
| 10. | 10.05 | p-Cymene | 1021 | 1020 | RI, MS | 0.4 | 0.6 |
| 11. | 10.18 | 1,8-Cineole | 1023 | 1026 | RI, MS, Co-I | 3.7 | 5.9 |
| 12. | 10.46 | (Z)-β-Ocimene | 1030 | 1032 | RI, MS | 3.0 | 2.9 |
| 13. | 10.84 | (E)-β-Ocimene | 1041 | 1044 | RI, MS | 2.3 | 3.0 |
| 14. | 11.29 | γ-Terpinene | 1054 | 1054 | RI, MS | 0.8 | 1.2 |
| 15. | 11.91 | *cis*-Sabinenehydrate | 1069 | 1065 | RI, MS | 0.8 | - |
| 16. | 12.41 | Terpinolene | 1083 | 1086 | RI, MS | tr | 0.6 |
| 17. | 12.55 | *trans*-Linalooloxide(furanoid) | 1086 | 1084 | RI, MS | 0.4 | 0.2 |
| 18. | 13.23 | Linalool | 1103 | 1095 | RI, MS, Co-I | 4.5 | 3.8 |
| 19. | 14.15 | *cis*-p-Menth-2-en-l-ol | 1126 | 1118 | RI, MS | 0.3 | tr |
| 20. | 15.50 | Sabinaketone | 1158 | 1154 | RI, MS | 0.6 | tr |
| 21. | 16.15 | Borneol | 1174 | 1165 | RI, MS | 1.7 | 1.5 |
| 22. | 16.57 | Terpinen-4-ol | 1184 | 1174 | RI, MS | 6.2 | 4.6 |
| 23. | 17.05 | Myrtenal | 1195 | 1195 | RI, MS | tr | - |
| 24. | 17.26 | α-Terpineol | 1200 | 1196 | RI, MS | 5.2 | 5.5 |
| 25. | 18.75 | Thymol, methylether | 1235 | 1232 | RI, MS | tr | tr |
| 26. | 20.32 | Geranial | 1272 | 1264 | RI, MS, Co-I | tr | tr |
| 27. | 20.96 | DihydroedulanI | 1288 | 1288 | RI, MS | 0.9 | 0.5 |
| 28. | 22.07 | (2E,4E)-Decadienol | 1314 | 1319 | RI, MS | tr | tr |
| 29. | 24.61 | α-Copaene | 1375 | 1374 | RI, MS | tr | tr |
| 30. | 25.02 | β-Bourbonene | 1385 | 1387 | RI, MS | 2.8 | 1.7 |
| 31. | 25.36 | β-Elemene | 1394 | 1389 | RI, MS | tr | tr |
| 32. | 26.50 | (E)-Caryophyllene | 1422 | 1417 | RI, MS | 8.4 | 8.5 |
| 33. | 26.87 | β-Copaene | 1431 | 1430 | RI, MS | 0.4 | 0.3 |
| 34. | 27.44 | Aromadendrene | 1445 | 1439 | RI, MS | tr | - |
| 35. | 27.92 | α-Humulene | 1457 | 1452 | RI, MS | 1.3 | 1.3 |
| 36. | 28.14 | Alloaromadendrene | 1463 | 1458 | RI, MS | 0.7 | 0.5 |
| 37. | 28.85 | γ-Muurolene | 1481 | 1478 | RI, MS | tr | tr |
| 38. | 29.09 | GermacreneD | 1485 | 1484 | RI, MS | 17.4 | 22.5 |
| 39. | 29.46 | epi-Cubebol | 1496 | 1493 | RI, MS | tr | tr |

**Table 5.** *Cont.*

| N°. | t ret., min | Compound | RIexp | RIlit | Method of Identification | c% | |
|-----|-------------|----------|-------|-------|--------------------------|-----|-----|
| | | | | | | Steam/Leaves | Flowers |
| 40. | 29.61 | Bicyclogermacrene | 1500 | 1500 | RI, MS | 2.4 | 1.9 |
| 41. | 29.78 | α-Muurolene | 1504 | 1500 | RI, MS | tr | tr |
| 42. | 30.03 | (E,E)-α-Farnesene | 1510 | 1505 | RI, MS | 3.4 | 5.5 |
| 43. | 30.40 | γ-Cadinene | 1520 | 1513 | RI, MS | tr | 0.4 |
| 44. | 30.67 | δ-Cadinene | 1528 | 1522 | RI, MS | 1.9 | 1.9 |
| 45. | 32.05 | Elemol | 1538 | 1548 | RI, MS | 2.0 | 1.4 |
| 46. | 33.19 | Caryophylleneoxide | 1592 | 1582 | RI, MS | 12.5 | 7.4 |
| 47. | 33.66 | Guaiol | 1606 | 1600 | RI, MS | 0.5 | - |
| 48. | 34.18 | HumuleneepoxideIl | 1618 | 1608 | RI, MS | 1.4 | 0.9 |
| 49. | 35.15 | β-Eudesmol | 1647 | 1649 | RI, MS | 0.6 | - |
| 50. | 35.58 | α-Muurolol | 1650 | 1640 | RI, MS | 1.8 | 1.5 |
| 51. | 36.11 | α-Cadinol | 1662 | 1652 | RI, MS | 4.3 | 3.5 |
| 52. | 37.25 | Amorpha-4,9-dien-2-ol | 1706 | 1700 | RI, MS | 1.3 | 0.9 |
| | | | | | Total identified | 100.0 | 99.8 |
| Grouped components (%) | | | | | | | |
| Monoterpene hydrocarbons (1–5,7,9,12–14,16) | | | | | | 8.8 | 14.0 |
| Oxygen–containing monoterpenes (11,15,17–24,26,27) | | | | | | 24.3 | 22.4 |
| Sesquiterpene hydrocarbons (29–38,40–44) | | | | | | 38.7 | 44.4 |
| Oxygenated sesquiterpenes (39,45–52) | | | | | | 24.4 | 16.1 |
| Aromatic compounds (10,25) | | | | | | 0.4 | 0.6 |
| Others (6,8,28) | | | | | | 3.4 | 2.3 |

Present only in steams/leaves: *cis*-Sabinenehydrate; Myrtenal; Aromadendrene; Guaiol; β-Eudesmol. Present only in flowers: Agarospirol; Neral.

A total of fifty-two components were identified from wild plants (steam/leaves), mainly sesquiterpene hydrocarbons (38.7%), oxygen-containing monoterpenes, and oxygenated sesquiterpenes (24.3 and 24.4%). The most dominant components of OES from wild plants' flowers are: germacrene D (17.4%), caryophylleneoxide (12.5%), and (E)-caryophyllene (6.2%). In addition to the mentioned components in the stems/leaves of wild oregano, there are also: terpinen-4-ol (6.2%), α-terpineol (5.2%), linalool (4.5%), α-cadinol (4.3%), and 1.8-cineole (3.7%), see Table 5.

A total of fifty components were identified from wild plant flowers, mainly sesquiterpene hydrocarbons (44.4%) and oxygen-containing monoterpenes (22.4%). The most dominant components of OES from wild plant flowers are: germacrene D (22.5%), (E)-caryophyllene (8.5%), and caryophylleneoxide (7.4%). In addition to the above-mentioned components, they are also represented on a larger scale as 1,8-cineole (5.9%), α-terpineol (5.5%), terpinen-4-ol (4.6%), linalool (3.8%), α-cadinol (3.5%), and (E)-β-ocimene (3%). The other compounds always reached amounts lower than 3% (Table 5).

It is very interesting to point out that some components of essential oils are present only in steam/leave part of wild oregano plants (cis-sabinene hydrate, myrtenal; aromadendrene, guaiol; β-eudesmol), while others are present only in flower agarospirol (0.5) and neral (tr).

Germacrene D and caryophyllene oxide are present in wild and cultivated oregano in all parts of the plant. In our previous research, total sesquiterpenes and oxygenated monoterpenes were the most abundant in oregano essential oils [17]. In wild-growing *O. vulgare* subsp. *hirtum* growing in Montenegro, the dominant component was carvacrol in

*O. vulgare* subsp. *vulgare*—germacrene D. [1]. Linalool and thymol are the main components of *Origanum vulgare* from the southern part of Albania, while oregano from the northern area has a high content of caryophyllene-oxide and β-pinene [32].The main compounds of oregano from Bosnia were carvacrol and p-cymene [42].

The composition of the essential oil of oregano depends on the light intensity, which is modified by shading nets. The essential oil of the oregano plant grown in nonshaded conditions mainly consists of 4-terpineol, γ-terpinene, carvacrol, and p-cymene. When plants are grown in shaded conditions, the essential oil mainly consists of γ-terpinene, 4-terpineol, carvacrol, and p-cymene [43].

Growing conditions, such as soil moisture and nitrogen fertilization, have a minor effect on the composition of oregano essential oils. A water deficit at the time of flowering can cause an increase in essential oil content [13]. The origin of the oregano plants is the most important factor for quantity and essential oil quality composition.

Biogenetic precursors γ-terpinene and p-cymene for thymol and carvacrol are the main compounds in oregano essential oils, but with great variability in the percentage depending on the geographical origin [44–46]. The major compounds in oregano from south-west Serbia were: sabinene, terpinen-4-ol, 1,8 cineole, γ-terpinene, and caryophyllene oxide [47], and it has a very similar composition to OEOs from Poland [41]. Armenian oregano consisted mainly of sesqui- and monoterpenes [48].

Oregano populations (*O. vulgare* L. subsp. *vulgare*) from northern regions and at higher altitudes are characterized by lower contents of volatile components compared to oregano populations from the southern hemisphere and at sea level, and are often composed of cymyl-compounds, bornane type (e.g., borneol, camphor, camphene), acyclic (mainly linalool and linalyl acetate), and sabinyl compounds (e.g., sabinene) with a larger contribution of sesquiterpenes [49–51]. Among the four chemotypes found in *O. vulgare*, the plant could be placed in the thymol chemotype and seems to be different from most of the chemotypes found in other parts of the world, including Turkey [49], Austria [50], Lithuania [51,52], and Italy [53].which have been reported to be β-caryophyllene or cymyl-sabinyltypes, rich in β-caryophyllene, sabinene, spathulenol, and germacrene D.

Regarding the previously published data on the chemical composition of *O. vulgare* L. subsp. *vulgare*, it seemed that despite being a poor-oil similar to other northern chemotypes, the Iranian *O. vulgare* L. subsp. *vulgare* may be categorized into a totally different chemotype, "thymol" with a higher percentage of monoterpenes and dominant oxygenated terpenoids [34].

Several studies have indicated that the chemical compositions of the EOs extracted from various parts of plants are different. The OEOs extracted from different parts of plants (such as flowers or leaves) have different components and properties [54].Transferring plants from spontaneous flora to a new agro-ecosystem and growing them under different conditions in a changing environment can result in modifications in plant growth, reproduction, and chemical composition [55].

Modified light by nets allows for the maximization of photosynthetic performance in oregano, mainly in the presence of favorable outdoor conditions or high cultivation density, to increase plant productivity and achieve constant yields and product quality. The interaction between temperature and light irradiance and/or spectral quality may modify and regulate flowering time in screen-house crop production. The flowering process involves complex biochemical, anatomical, and morphological changes, which are synthetically described by four events: flower induction, flower evocation and initiation, and flower development. Flower induction consists of endogenous or exogenous signals determining changes in the plant's developmental program. In response to these, a chemical stimulus is transmitted to the meristematic apex, which is altered to produce flowers instead of leaves in a process called floral evocation. This is followed by the formation of flower buds, defined as flower initiation, and by flower or inflorescence development. Flowers from wild oregano have the strongest antioxidant activity in relation to cultivated oregano as well as in relation to the plant part (leaf/stem).

### 3.4. Antioxidant Activity

The antioxidant activity of oregano comes from phenolic compounds and their hydroxy groups. Due to the presence of nonvolatile components, oregano is one of the most commonly used aromatic plants. Differences in antioxidant capacity between cultivated plants (shaded and nonshaded) and wild oregano, as well as between individual parts of the plant, are significant.

There were no significant differences in the antioxidant capacity (AOX) of OEOs from the leaves/stem of cultivated oregano based on the cultivation method, but shaded plants exhibited slightly higher antioxidant capacity ($EC_{50}$—7.91 mg/mL) than nonshaded plants ($EC_{50}$—8.59 mg/mL). The $EC_{50}$ value in the flowers of cultivated oregano (24.63 mg/mL) is higher than that in the flowers of the wild ecotype, which means that antioxidant capacity was stronger in flowers from wild plants (4.78 mg/mL), see Table 6.

**Table 6.** $EC_{50}$ values of essential oil from the different origin and plants parts of oregano.

| Essential Oil | $EC_{50}$, mg/mL | | | |
|---|---|---|---|---|
| | Incubaton Time | | | |
| | Without Incubation | 20 min Incubation | 40 min Incubation | 60 min Incubation |
| Nonshaded oregano (stems and leaves) | / | | | 8.59 ± 0.034 |
| Shaded oregano (stems and leaves) | / | | | 7.91 ± 0.015 |
| Nonshaded oregano (flowers) | / | | | 24.63 ± 0.865 |
| Shaded oregano (flowers) | / | / | / | * |
| Wild oregano (stems and leaves) | / | / | / | * |
| Wild oregano(flowers) | / | | | 4.78 ± 0.052 |

* analyses were not performed due to a lack of samples.

EOs from different origins and plant parts of oregano and $EC_{50}$ values after 60 min of incubation are presented in Figures 2–5.

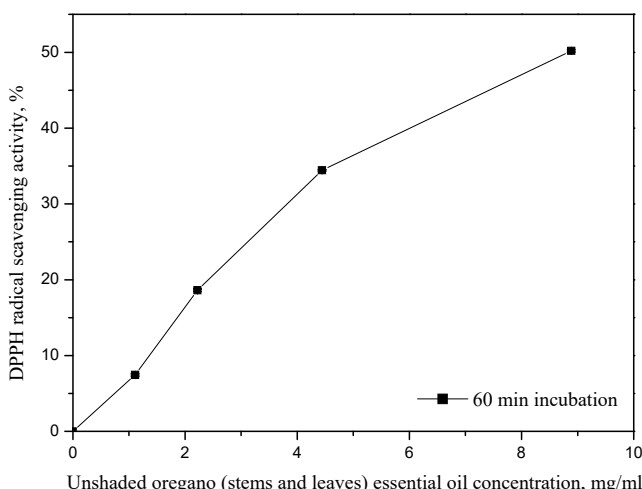

**Figure 2.** Antioxidant activity of nonshaded oregano (stems and leaves) essential oil.

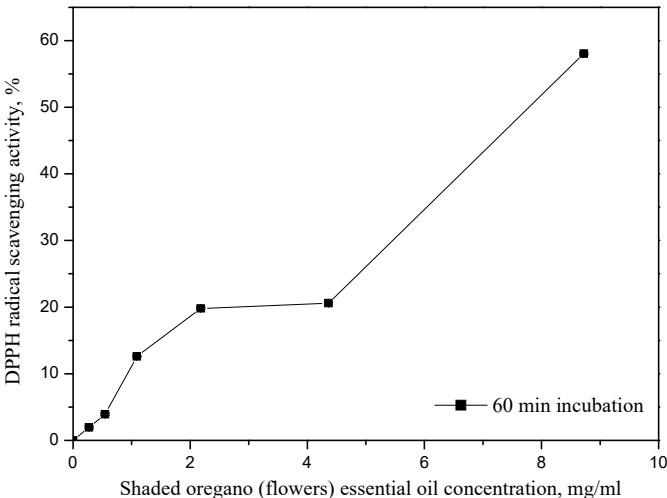

**Figure 3.** Antioxidant activity of shaded oregano (flowers) essential oil.

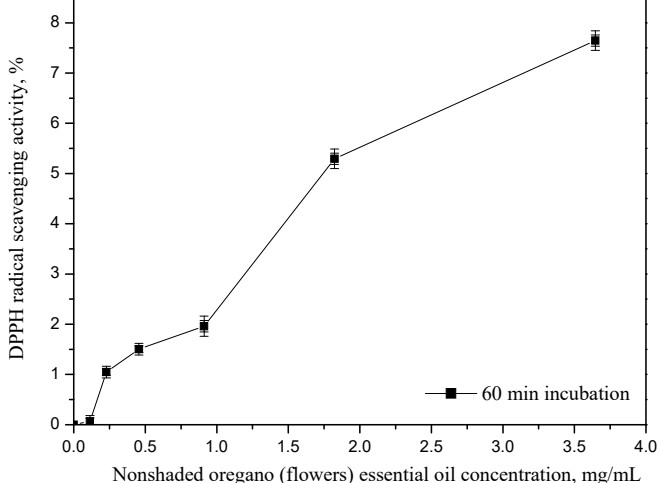

**Figure 4.** Antioxidant activity of nonshaded oregano (flowers) essential oil.

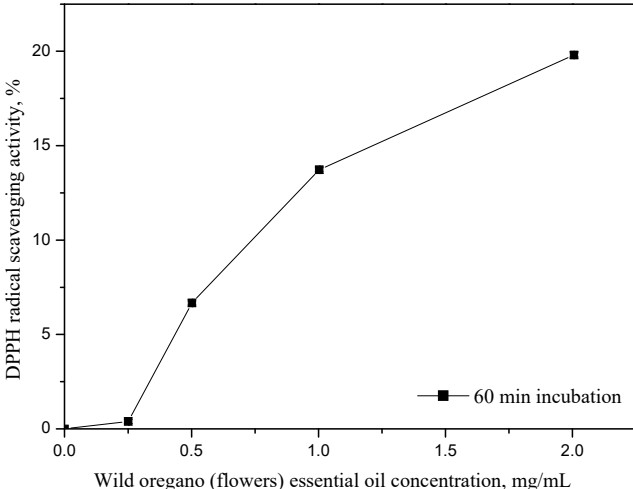

**Figure 5.** Antioxidant activity of wild oregano (flowers) essential oil.

EOs samples from shaded thyme, marjoram, and oregano plants showed higher antioxidant activity than nonshaded plants [17]. In our previous experiment, thyme EOs were reported to be the best antioxidants in comparison to oregano and marjoram [17].

Shade nets provide biosynthesis for components with antioxidant properties [15]. Our results with the covering of oregano plants confirm the findings of Ilić et al. [24], who reported that lemon balm and mint in full sunlight showed the lowest levels of EOs and a lower antioxidant activity. Shaded basil plants showed higher antioxidant activity than nonshaded basil plants. These results are in agreement with the results of other authors [16,17]. However, in our study, the $EC_{50}$ values of the oregano essential oils were slightly different from those identified in other works possibly because of the various oregano plant origins or differences in cultivated oregano (cover plants with nets).

Differences in antioxidant activity are evident between cultivated and wild oregano. $EC_{50}$ values were higher in wild oregano compared to cultivated oregano, which means that stronger antioxidant activity was recorded in cultivated oregano [55]. Differences in antioxidant activity exist in different parts of plants [56]. Thus, antioxidant activity is similar in flower, leaves and roots, while it is significantly lower in the stem. *Origanum* species from southwestern Serbia exhibited strong antioxidant activity with $EC_{50}$ values between 34.5 and 86 mg/mL [47].

The essential oil of Iranian chemotype "thymol" demonstrated radical scavenging ability with an $EC_{50}$ of 2.5 μg/mL [34]. The comparison of antioxidant activity results published in many papers is difficult to standardize because the data is significantly influenced by the method of extraction and the analytical method used for their determination.

The harvesting time might also influence the oregano's antioxidant activity (AOX) in EOs obtained from the leaves and flowers of *O. glandulosum* [57]. The OEOs samples obtained in June showed lower $EC_{50}$ values and higher AOX capacity than those obtained in the following months during the summer [58]. Essential oils of oregano (EOOs) are very complex mixtures of compounds, in which the major constituents are terpenes, generally mono- and sesquiterpenes. The principal terpenes identified in the different species of oregano are carvacrol, thymol, γ-terpinene, and p-cymene; while terpinen-4-ol, linalool, β-myrcene, *trans*-sabinene hydrate, and β-caryophyllene are also present. Different studies have shown that the AOX effects of EOO are related to the presence of phenolic structures, such as thymol and carvacrol, and so these compounds could replace the synthetic antioxidants currently used in the food industry, and due to their natural origin, they can also improve health.

Differences in the AOX activity and chemical composition of EOO have been reported in relation to the geographic origin, the mode of extraction, or plant differences in the phenological stage. The AOX properties of EOs are thought to be related, through mechanisms, such as (1) free radical scavenging activity, (2) modulation of AOX enzymes (superoxide dismutase), and (3) inhibition of pro-oxidation.

As effective antioxidants, OEOs are widely used in food products as an integral part, but also as edible films or edible coatings in the packaging of fresh products [59]. In addition to its fresh use, oregano is increasingly used as a raw material for the production of essential oils, flavorings, and food additives, which creates higher profits by developing small farms and obtaining raw materials for local agro food industries.

## 4. Conclusions

In addition to its medicinal properties, oregano also has great economic importance as a spice plant and a source of essential oils that can be used for various purposes. The eco-geographical characteristics of Serbia represent good conditions for the spontaneous growth of *Origanum vulgare* subsp. *vulgare* L. plants in many places throughout the country. Oregano is also being cultivated, but in a limited area. The future recommendations would be based on the collection of wild and cultivated oregano (using adequate agricultural techniques with crop shading and adequate plant density) and the goal of obtaining an increased amount of a high-quality essential oil. The limitations of the study would refer to wild oregano and the impossibility of changing anything important in its expansion, only about the importance of rational use-collection. The practical significance of this study should provide the possibility of obtaining a higher yield and better quality of EOs from

wild and cultivated oregano in order to increase their application in the food industry and pharmacology. Optimized production techniques using plant shading from the present study could provide useful methods for improving the content and composition of oregano essential oils.

**Author Contributions:** Z.I. and L.S. were heads of the research group, planned the research, analyzed it, and wrote the manuscript; L.M. and L.Š. conducted the experiment in the field; and J.S., A.M. and D.C. performed analyses on physical properties and chemical composition in the laboratory. All authors have read and agreed to the published version of the manuscript.

**Funding:** This research received external funding from a program for financing scientific research work, with grant numbers 451-03-68/2022-14/200133 and 451-03-68/2022-14/200189 was financially supported by the Ministry of Education Science and Technological Development of the Republic of Serbia.

**Institutional Review Board Statement:** Not Applicable.

**Informed Consent Statement:** Not Applicable.

**Data Availability Statement:** All the data is available in the manuscript file.

**Conflicts of Interest:** The authors declare no conflict of interest. The funders had no role in the design of the study; in the collection, analysis, or interpretation of data; in the writing of the manuscript or in the decision to publish the results.

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
