# Peer review of "The Yield, Chemical Composition, and Antioxidant Activities of Essential Oils from Different Plant Parts of the Wild and Cultivated Oregano (Origanum vulgare L.)"

_horticulturae, doi:10.3390/horticulturae8111042_

Round 1

Reviewer 1 Report

The MS is of great interest and presents easily transferable results to the agri-food sector.

Author Response

Thanks for you support 

Reviewer 2 Report

In this manuscript, the authors tested the essential oil yields of oregano plants grown under pearl nets with a shade index of 40%, cultivated in shade-free conditions, and the wild oregano. Then they compared the chemical composition and antioxidant activity capacity of essential oils extracted from different parts of these three plants (flowers or leaves/stems). The authors stated that plant shading technology can achieve high and quality oregano yields and better-quality parameters in terms of specific OEOs components and meet the different requirements of market and industrial sectors.

The logic of the introduction part was weak. The readers did not know why the authors want to detect the essential oil yields of oregano from different conditions. And the current study of this area was not explained sufficiently.

In section 2.4, it was said that the antioxidant activity (DPPH assay) was measured after 20 min of incubation at room temperature. However, in Figure 7-10, it was shown as “60 min incubation”. What did this difference mean?

In Table 6, the data for shaded oregano (flowers) and wild oregano (stems and leaves) were missing, but these data were critical for the completeness and reasonableness of this study.

The authors showed the structures of some components, but these structures did not show any help for readers to understand this article.

Author Response

Please open Attach 

Reviewer 3 Report

The article " The yield, chemical composition, and antioxidant activities of essential oils from different plant parts of the wild and culti-vated oregano (Origanum vulgare L.)" is well-designed study, which is very interesting but with several grammatical mistakes. This article can be reconsidered after revising the following comments.

Comment # 1: Replace “ativities which are associated with the presence of their phenolic components” with “ativities, which are associated with the presence of their phenolic components.

Comment # 2: Mention the external factors “factors of the external environment [14]?

Comment # 3: Authors should stress on limitations why this study is important in the introduction part.

Comment # 4: What could be the practical significance of this study add with the objectives of this study?

Comment # 5: Authors should compare the antioxidants profile with other species having the same characteristics and explain the superiority of oregano.

Comment # 6: Provide the explanation? What chemical process lead the difference in the flower composition during shedding and open air growth and cultivation?

Comment # 7: The conclusion part should be added with future recommendations, limitations of study and practical significance.

Author Response

Please open Atach

Round 2

Reviewer 2 Report

This manuscript has been greatly improved and now can be accepted by horticulturae.

Author Response

Please open Attach and recognised our contribution and new version R2
